# New Insights in Microbial Species Predicting Lung Function Decline in CF: Lessons from the MucoFong Project

**DOI:** 10.3390/jcm10163725

**Published:** 2021-08-21

**Authors:** Florence Francis, Raphael Enaud, Perrine Soret, Florian Lussac-Sorton, Marta Avalos-Fernandez, Stéphanie Bui, Michael Fayon, Rodolphe Thiébaut, Laurence Delhaes

**Affiliations:** 1CHU de Bordeaux, Department of Public Health, F-33000 Bordeaux, France; florence.francis@u-bordeaux.fr (F.F.); rodolphe.thiebaut@u-bordeaux.fr (R.T.); 2Bordeaux Population Health Research Center, Univ. Bordeaux, Inserm, UMR 1219, F-33000 Bordeaux, France; perrine.soret@gmail.com (P.S.); marta.avalos-fernandez@u-bordeaux.fr (M.A.-F.); 3Centre de Recherche Cardio-Thoracique de Bordeaux, Univ. Bordeaux, U1045, F-33000 Bordeaux, France; raphael.enaud@chu-bordeaux.fr (R.E.); florian.lussac-sorton@u-bordeaux.fr (F.L.-S.); stephanie.bui@chu-bordeaux.fr (S.B.); michael.fayon@chu-bordeaux.fr (M.F.); 4CHU de Bordeaux, Univ. Bordeaux, FHU ACRONIM, F-33000 Bordeaux, France; 5CHU de Bordeaux, CRCM Pédiatrique, CIC 1401, F-33000 Bordeaux, France; 6INRIA SISTM Team, F-33405 Talence, France; 7Laboratoire Servier, 50 Rue Carnot, 92284 Suresnes, France; 8CHU de Bordeaux, Service de Parasitologie-Mycologie, F-33000 Bordeaux, France; 9The Mucofong Investigation Group, 28 rue Roger Amsler, CEDEX 01, 49045 Angers, France; jean-philippe.bouchara@univ-angers.fr

**Keywords:** cystic fibrosis, longitudinal analysis, *Aspergillus fumigatus*, *Candida albicans*, mycobiota

## Abstract

Several predictive models have been proposed to understand the microbial risk factors associated with cystic fibrosis (CF) progression. Very few have integrated fungal airways colonisation, which is increasingly recognized as a key player regarding CF progression. To assess the association between the percent predicted forced expiratory volume in 1 s (ppFEV1) change and the fungi or bacteria identified in the sputum, 299 CF patients from the “MucoFong” project were included and followed-up with over two years. The relationship between the microorganisms identified in the sputum and ppFEV1 course of patients was longitudinally analysed. An adjusted linear mixed model analysis was performed to evaluate the effect of a transient or chronic bacterial and/or fungal colonisation at inclusion on the ppFEV1 change over a two-year period. *Pseudomonas aeruginosa*, *Achromobacter xylosoxidans*, *Stenotrophomonas maltophilia*, and *Candida albicans* were associated with a significant ppFEV1 decrease. No significant association was found with other fungal colonisations. In addition, the ppFEV1 outcome in our model was 11.26% lower in patients presenting with a transient colonisation with non-pneumoniae *Streptococcus* species compared to other patients. These results confirm recently published data and provide new insights into bacterial and fungal colonisation as key factors for the assessment of lung function decline in CF patients.

## 1. Introduction

Cystic fibrosis (CF) is the most common inherited disease in Caucasian populations, affecting about 70,000 individuals in the world [1]. This multisystem disease affects primarily the lungs, causing repeated lung infections and chronic lung inflammation which lead to progressive respiratory failures associated with significant morbidity and early death.

Several predictive models have been proposed to identify risk factors associated with severe lung function decline, morbidity, or mortality in CF [2,3,4,5,6,7,8,9,10,11,12,13,14,15]. These models underscored the importance of several clinical factors, including microbial airway colonisations, when assessing morbidity and mortality in CF. However, none but two of these studies [4,12] have taken into consideration fungal flora (mycobiota) of the CF airways. These two studies were focused on managing spontaneous pneumothorax and scoring the risk of death or lung transplantation (LT) in CF, respectively [4,12], and provided data and scores that might be useful for daily practice but need to be used by other clinicians from other CF centres. In addition, fungi are increasingly recognized as key players in the natural course of CF lung disease and as real threats in the case of lung transplantation (LT) [1,4,12,16,17,18].

In the present study, our aim was to determine whether bacterial and/or fungal colonisations of the airways were independently associated with CF patients’ lung function progression in the French national multicentre longitudinal “MucoFong” study.

## 2. Materials and Methods

### 2.1. Study Design and Population

Data collected from the prospective MucoFong project, which was supported by the French National Clinical Research program (“PHRC” protocol number 06/1902, 2007–2012 [18,19]), were used in the present study. The main overall aim of MucoFong was to set-up a standardized mycological analysis of sputa and to estimate fungal prevalence rates in CF, which led to an in-depth analysis of sputa.

### 2.2. Ethics

This study was approved by the Institutional Ethics Committees of Lille University Hospital (CPP Number: 06/84); written informed consent was provided by all participants [18,19].

Clinical data including standard spirometry, therapeutic, radiological, and biological data along with allergic bronchopulmonary aspergillosis (ABPA) criteria were prospectively collected at each visit. CF participants underwent 3 visits during the 2 years of the follow-up [19].

### 2.3. Microbial Analysis

Bacterial cultures were performed in agreement with the European CF Society guidelines. Briefly, in agreement with the French guidelines, 10 µL of pure sputum sample were plated on a selective agar for Gram-negative bacilli isolation (such as MacConkey type), a selective agar for *Pseudomonas aeruginosa* isolation (optional if the previous selective agar allows accurate identification of *P. aeruginosa* colonies), a selective agar for *Burkholderia cepacia* complex isolation (such as PC or BCSA), and a selective agar for *Staphylococcus aureus* isolation including small colony variants (e.g., hypersaline mannitol agar, chromogenic media). These media were incubated at 35 ± 2 °C in an aerobic atmosphere for selective media and in a CO_2_-enriched atmosphere for non-selective media. A minimum detection limit equal to 10^2^ CFU/mL were used to interpret growth on each type of media. In parallel, 10 µL of sample diluted to 1/1000 (detection threshold equal to 10^5^ CFU/mL) were plated on an agar allowing the growth of *Streptococcus pneumoniae* and *S. aureus* with inhibition of Gram-negative bacteria (e.g., fresh blood ANC, CAP), and a cooked blood agar with or without bacitracin and with growth factors for the cultivation of *Haemophilus influenzae*. These two media were incubated at 35 ± 2 °C in a CO_2_-enriched atmosphere. Fungal growth was detected using 6 semi-selective media according to the same standardized protocol performed by all the participating centres as previously described [19].

Given their clinical relevance [8,16,20,21,22], the presence of the mucoid phenotype of *P. aeruginosa* isolates, and isolates of *S. aureus* with methicillin resistance (MRSA) or not (MSSA), *B. cepacia* complex, *Achromobacter xylosoxidans*, *Stenotrophomonas maltophilia*, non-tuberculous mycobacteria, or *Streptococcus* species was recorded. If bacteria were recorded at least once during the year prior to the visit without meeting the criteria for chronic colonisation, bacterial colonisation was considered transient. Chronic colonisation with *P. aeruginosa* (and by extension with either *S. aureus*, *S. maltophilia*, *B. cepacia* complex, and *A. xylosoxidans*) was defined as the presence of bacterial isolates in 3 consecutive cultures with at least 1 month between the positive cultures, during the previous 6 months [20].

Chronic colonisation with *Aspergillus fumigatus* (and by extension with other fungal species) was deemed present when fungi were isolated in 2 consecutive cultures with at least 1 month between the positive cultures during the previous 12 months [23]. Transient fungal colonisation was defined as the presence of a given micromycete in one sample only, either at the inclusion visit or during the year prior to inclusion.

### 2.4. Outcome Assessment

Given its major role in clinical practice and as it is currently the gold standard measure of disease severity, percent predicted forced expiratory volume in 1 s (ppFEV1) was selected as the main outcome variable. Spirometry was carried out in each centre, for routine care, by approved services applying the recommendations of the American Thoracic Society and European Respiratory Society in force [24]. The median of this parameter (ppFEV1) was modelled to assess the relationship between microbial colonisations at inclusion and lung disease progression over the 2 years following inclusion.

### 2.5. Prognostic Score Assessment

The recently proposed risk score for the prediction of within 3-year death or lung transplantation (LT) [12] was estimated for each patient at inclusion. Based on the 8 predictors proposed (i.e., ppFEV1, body mass index (BMI), *B. cepacia* complex colonisation, number of intravenous antibiotics courses per year, number of days of hospitalization per year, oral corticosteroids therapy, long-term oxygen therapy, and non-invasive ventilation), a weighted continuous score was individually calculated, namely, a high prognostic score corresponding to a high risk of LT or death.

### 2.6. Statistical Analysis

Analyses were performed using the statistical analysis software SAS version 9.4 (SAS Institute, Cary, NC, USA). Descriptive analyses have been performed to summarize the CF population at inclusion and during the follow-up. Continuous variables were presented as mean with standard deviation (Mean ± SD) for normal distribution or median with interquartile range (25th–75th percentile, IQR) for non-normal distribution. Qualitative variables were presented as *n* (%).

As data have been collected repeatedly during the follow-up, a linear mixed model analysis was used to evaluate the effect of transient or chronic colonisations with fungi and/or bacteria at the inclusion on the changes of ppFEV1. The model included a random intercept; parameters were estimated with the maximum likelihood method. A graphical analysis of the data revealed a heterogeneous change of ppFEV1 values among patients throughout the follow-up. It was therefore decided to consider time as a discrete variable (one variable per visit). No additional time interaction was required in the model; therefore, the effects of bacteria or fungi were modelled as an average difference of ppFEV1 from the inclusion onward.

The analyses were adjusted for potential confounding factors (study centre, age, sex, BMI, ABPA, and gastroesophageal reflux). According to the unadjusted univariate analyses, potential microbial predictors of the outcome such as transient or chronic bacterial and fungal colonisations (yes/no) were initially introduced in the model. A univariate analysis compared ppFEV1 from patients infected/colonised with a given organism to all other patients. Parameters with *p* ≤ 0.25 in unadjusted analyses were considered for inclusion in the multivariable model. Then, a stepwise procedure was performed in multivariable adjusted analyses to select the subset of variables independently associated with the outcome (significant threshold at 0.05).

Accordingly, an initial model including all selected variables at an unadjusted step was performed. Then, at each step, before a variable was selected to be removed from the model, a hypothesis test was performed in order to state whether its contribution was significant or not. The procedure ended when all non-significant variables were removed. Estimate fixed effects for significant parameters were reported as β coefficient, confidence interval 95% (95% CI), and *p*-value. The β coefficient represents the mean difference of the ppFEV1 change between, respectively, patients with a chronic or transient colonisation at inclusion and those without colonisation adjusted on other variables. A *T*-test or partial Fischer test was used to assess whether the β coefficient was significantly different from zero. Residual plots (marginal and conditional) were analysed to check their normality and homoskedasticity.

Given the polymicrobial constitution of CF airways flora, additional analyses were implemented to search for potential interactions between microorganisms. 

### 2.7. Model Robustness

To assess whether the microorganisms identified as associated with a poor ppFEV1 level in the main analysis were also associated with worse prognostic scores, an adjusted linear regression was performed using the prognostic score recently developed [12] as the outcome variable.

## 3. Results

### 3.1. Patients’ Characteristics

A total of 299 patients were included in the study, of whom 237 were monitored at each visit during the 2 years of follow-up (Figure 1).

Patients’ characteristics, lung function results, microbial cultures of sputum, and therapeutics at study entrance are summarized in Table 1. Patients were adults, 23.7 (±10.9) years old; 143 (47.8%) were female; 255 (85.3%) had at least one F508del mutation of the *CFTR* gene; and the majority had exocrine pancreatic dysfunction (78.6%). The median [IQR] Shwachman–Kulczycki score (S-K score, [2]) was 74.6 [IQR: 65–90].

Most patients were colonised with *P. aeruginosa*, *S. aureus*, *Haemophilus influenzae*, and non-pneumoniae *Streptococcus* species (including *S. mitis* group species; *n* = 19, 6.5%; Table 1), and to a lesser degree with *S. maltophilia*, *A. xylosoxidans*, *B. cepacia* complex species and/or non-tuberculous mycobacteria (Table 1). *Candida albicans* (45.5%) and *A. fumigatus* (26.8%) were the two major fungal species at inclusion, whereas *Aspergillus flavus* and *Scedosporium* species were isolated, respectively, in 3% and 4% of patients (Table 1). As previously described [1,19], the number of fungal species isolated increased with patient age: from none (18.9%) to one (42%), two (29.6%), three (8.2%), four (0.8%), and five (0.4%) species. In addition, patients colonised with *P. aeruginosa* were mainly colonised with two or more microbial species. Indeed, 33 of the 75 patients colonised with non-mucoid *P. aeruginosa* were co-infected with only one of the following species: mucoid *P. aeruginosa (n =* 7), non-pneumoniae *Streptococcus* (*n* = 4), MSSA (*n* = 11), MRSA (*n* = 2), *C. albicans* (*n* = 4), or *A. fumigatus* (*n* = 5) species. Furthermore, 29 of the 77 patients colonized with *P. aeruginosa* (mucoid strains) were co-infected with only one of the following species: non-mucoid *P. aeruginosa (n =* 7), non-pneumoniae *Streptococcus* (*n* = 2), MSSA (*n* = 3), MRSA (*n* = 1), *C. albicans* (*n* = 7), or *A. fumigatus* (*n* = 9) species. Transient and chronic colonisations with *P. aeruginosa*, *S. aureus*, *C. albicans*, or *A. fumigatus* were mainly observed at baseline (Figure 2a,b). Less than 5% were chronically colonised with *S. maltophilia*, *A. xylosoxidans*, *A. flavus*, or with the *Scedosporium* species (Figure 2b). Transient colonisation with non-pneumoniae *Streptococcus* and with *C. albicans* were observed in 9.2% and 26.4% of cases (Figure 2a), with a chronic colonisation with *C. albicans* being stated in 31.8% (Figure 2b).

Regarding medications at inclusion, most of the patients received rhDNAse (170/299; 59.5%), inhaled antibiotics (86/299; 28.8%), and/or inhaled steroids (168/299; 56.2%). While oral antibiotics, in particular ciprofloxacin formulations, remained the most often prescribed antimicrobial agents (74.9%), intravenous antibiotics were prescribed in 153 cases (51.2%) (Table 1). With respect to ABPA management, antifungal drugs were prescribed during the previous 6 months in 61 patients (20.4%) (Table 1).

### 3.2. ppFEV1 Characteristics and Univariate Analysis

Medians of ppFEV1 were 67% [43–91%] (*n* = 295 patients), 70% [41–92%] (*n* = 254 patients), and 69% [47–91%] (*n* = 231 patients) at inclusion, respectively, the first and second years of follow-up (Table 2). 

At baseline, ppFEV1 were lower in the case of a transient colonisation with non-pneumoniae *Streptococcus* species (Figure 2a), or chronic colonisation with *P. aeruginosa*, *S. maltophilia*, *B. cepacia* complex, *A. xylosoxidans*, *C. albicans*, *A. flavus*, and *Candida glabrata* (Figure 2b). An unadjusted analysis identified six microbial variables associated with a steeper decline of the ppFEV1 level compared to patients with no reported colonisation (Table 3). Among them, chronic infection with *P. aeruginosa* but also transient colonisation with *P. aeruginosa* (mucoid and non-mucoid strains) or colonisation with *C. albicans* were the most significantly associated with a decrease in the ppFEV1 level. Given the polymicrobial composition of the bacterial and fungal airway communities, numerous coinfections (non-mucoid or mucoid strains of *P. aeruginosa* in combination with either *S. aureus*, non-pneumoniae *Streptococcus* species, *A. fumigatus*, *Scedosporium sp*., *C. albicans*, or *C. glabrata)* were explored using a univariate analysis and did not reveal any significant impact on ppFEV1 (data not shown).

### 3.3. Bacterial and Fungal Colonisations Predicted ppFEV1 Level

After the stepwise selection, colonisation with *S. pneumoniae*, non-pneumoniae *Streptococcus* species, *A. xylosoxidans*, *S. maltophilia*, mucoid and non-mucoid strains of *P. aeruginosa*, and *C. albicans* were included in the multivariable model (Table 4).

Transient as well as chronic colonisations with *S. maltophilia*, *P. aeruginosa*, and *C. albicans* were significantly and independently associated with ppFEV1 deterioration during the two years of follow-up. Furthermore, transient colonisation with non-pneumoniae Streptococcus was associated with a 13.6% decrease in ppFEV1 (β = −13.62, 95%CI: −25.12; −2.12, *p* = 0.02). The effect of *A. xylosoxidans* and *S. pneumoniae* colonisation did not reach the significant threshold of 0.05. The positive association between *S. pneumoniae* colonisation and the ppFEV1 level identified in the univariate analysis was confirmed in the multivariate model (Table 4).

A fungal colonisation was associated with a decrease of ppFEV1 (β = −12.44, 95%CI: −18.87; −6.02 and β = −6.33, 95%CI: −12.68; 0.01) for chronic and transient *C. albicans* colonisation, respectively).

Given the limited number of patients with chronic colonisation by *B. cepacia* species complex (*n* = 3), *Mycobacterium avium complex (n =* 4), *Mycobacterium abscessus complex (n =* 5), *A. flavus* (*n* = 9), *C. glabrata* (*n* = 3), and other *Candida* species including *C. parapsilosis* (*n* = 3), these microorganisms have not been taken into account in our mixed model. Results were adjusted based on medical centre, age, sex, BMI, presence of ABPA, and/or gastroesophageal reflux disease.

### 3.4. Assessment of the Performance of our Mixed Model Using the Prognostic Score

With the exception of *C. albicans* and non-pneumoniae *Streptococcus* spp. colonisations, transient and chronic colonisations with *P. aeruginosa*, *A. xylosoxidans*, and *S. maltophilia* were significantly associated with worse adjusted 3-years prognostic scores of death or LT [12] at inclusion (Table 5), confirming the suitable performance of our mixed model.

Results were adjusted based on medical centre, age, sex, BMI, presence of an ABPA, and/or a gastroesophageal reflux disease.

## 4. Discussion

In the present work, we used microbial data from the most recent prospective multicentre cohort of French CF patients (MucoFong), based on a standardized sputum assessment protocol [19]. A linear mixed model was developed using a wide variety of microbial variables including fungi. It aimed to identify microbial species of ppFEV1 decline, with a statistical model that allowed patients to have varying baseline ppFEV1 measurements and different rates of change in ppFEV1 over time, as recently proposed [25].

Our results were in agreement with previous findings indicating that several concomitant chronic bacterial colonisations at the time of diagnosis (*P. aeruginosa, B. cepacia*, and *S. maltophilia*) confer a worse ppFEV1 level in CF. Traditional CF pathogens remain the best predictors of disease outcomes in adults with CF [3,5,6,10,12,14,16,17,26,27,28,29,30,31,32,33]. Transient bacterial colonisations, especially non-pneumoniae *Streptococcus* colonisation, were also factors associated with a worse level of ppFEV1 in our population. *C. albicans* colonisation was also associated with a worse level of ppFEV1. This last result is congruent with the principal component analysis in which *C. albicans* was more present in sputa from patients with severe CF disease [16]. It also supports results which evidenced that *C. albicans* colonisation presages FEV1 decline [30,34], although the clinical relevance of yeast colonisation remains a matter of debate in CF.

Transient colonisation with *S. pneumoniae* at inclusion appeared to be associated with a favourable ppFEV1 change, in agreement with a previous report showing that *S. pneumoniae* carriage is surprisingly high but not associated with more severe CF disease [35]. On the other hand, non-pneumoniae *Streptococcus* carriage was correlated with decreased ppFEV1 in one of the three models analysed in our population. Several studies reported the pathogenic potential of species such as *Streptococcus anginosus* or *mitis* group in CF [36,37]. Moreover, a longitudinal analysis of the bacterial composition and dynamics of sputum samples from a CF adult patient chronically colonised by *P. aeruginosa* revealed the key role of the *Streptococcus milleri* group in establishing chronic lung disease but also in exacerbation initiations [38]. These data highlight that the mechanisms of pathogenicity are complex and rely on the polymicrobial context of CF airways, in which *P. aeruginosa* is a recognized major pathogen interacting with the other members of the CF lung microbial community including non-pneumoniae *Streptococcus*, which may shape *P. aeruginosa* pathogenicity [38,39,40,41].

Several potential predictors, in particular colonisation with MSSA, MRSA, and *A. fumigatus*, were not shown to have a significant impact on the primary outcome (ppFEV1 average change during the 2 years of follow-up) in our model, in contrast to previous results [28,29,31,32,42]. However, MRSA prevalence appears to vary according to age and geographical situation in CF [14,28,42]. Regarding colonisation by moulds, the limited number of cases with *Scedosporium* species (*n* = 10) recorded in the present study may explain the lack of power to reach a significant threshold in the final analysis. Among our CF population, 80 patients (26.8%) had *A. fumigatus* positive cultures at inclusion, including 57 with a chronic colonisation. In agreement with previous studies, we did not find a significant association between *A. fumigatus* colonisation and worse lung function [12,17]. However, Nkam et al. [12] showed that CF patients who died or underwent lung transplantation were more likely to have been diagnosed with ABPA, and there was a similar prevalence of ABPA between Nkam et al.’s population and ours (18.9%, Table 1 vs. 23.1% [12]). Both ABPA and *Aspergillus* sp. carriage have been associated with impaired lung function [30]. Conversely, the relationship between chronic colonisation with *A. fumigatus* and lung function recently evaluated using a linear mixed model appeared to exhibit a complex two-way relationship in CF [43].

Taken together, these results highlight (i) the potential pathogenicity in CF of species derived from the oral cavity and usually considered as clinically non-significant, and (ii) the polymicrobial signature of CF airway colonisation which may contribute as a collective entity to lung function decline [36,44].

Our study has some limitations. The study targeted an adult population able to produce sputum samples (age median of 22.3 years; CI95%: 16.7–29.9) leading to a selected population with more severe CF disease. The main outcome variable was ppFEV1, due to its major role in clinical practice and how it is currently the gold standard measure of disease severity. However, several recent parameters have emerged as promising alternative outcome variables to measure lung progression. Besides cross-sectional imaging by chest computed tomography or magnetic resonance imaging, the lung clearance index or multiple breath washout were shown to be more sensitive compared to ppFEV1, FVC, or to FEF25-75 [45,46,47]. Unfortunately, the study design did not allow us to identify non-pneumoniae *Streptococci* at the species level nor to quantify each microbial species since the MucoFong project was a multicentre study focused on identification and prevalence of fungi [18,19]. So far, the demographic, clinical, biological, and therapeutic characteristics of our population have been similar to published data [3,12]. In the last decade, linear and nonlinear models have emerged and demonstrated the clinical relevance in modelling pulmonary exacerbation or spirometry data in CF [25,30,32,43,48]. 

These models identified several bacterial colonisations as factors associated with the ppFEV1 level, but few of them have addressed the mycobiota effect [17,30,43]. Two additional models scoring LT, mortality, and morbidity included the mycobiota [4,12]. Yet, estimating CF life expectancy, morbidity, mortality, and scoring the risk of LT or death are central but late outcomes, known to depend heavily on the CF patients’ characteristics and their access to medical care and medications [44]. Additionally, prognostic factors have changed over time with the growing number of adults with CF, imposing a renewal in assessing lung function progression and microbial colonisation which may take advantage from next generation sequencing to simultaneously analyse the whole microbial community (bacteria, fungi, and viruses) without a priori knowledge of existing microorganisms [18,39]. 

To conclude, CF prognosis has dramatically improved over the past decades leading to an urgent need to reappraise the prognostic factors of CF progression, e.g., routinely assessing fungal colonisation and/or infection that have emerged as a new concern in CF [1,30,34,43]. Our model combining both items confirms the complex nature of CF lung disease, and provides new insights into bacterial and fungal colonisations as key factors in assessing and predicting lung function progression in CF. It emphasizes the need for regularly monitoring the bacterial and fungal colonisations of CF in lungs by using both culturing methods that allow microorganism phenotyping (such as in vitro sensitivity to antimicrobial agents) and molecular tools such as qPCR [49]. It also emphasizes the need, in the near future, for metataxonomics that provide the opportunity to identify all microbial species (including those difficult to culture) [18,39,41].

## Figures and Tables

**Figure 1 jcm-10-03725-f001:**
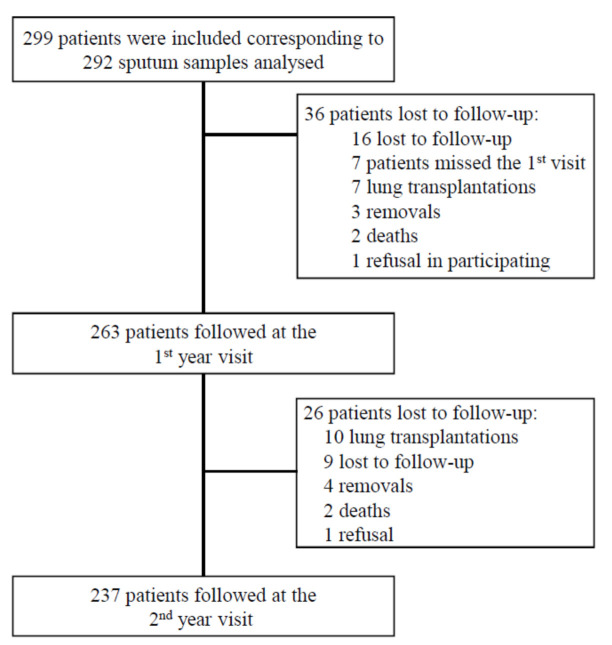
Flowchart showing inclusion and follow-up process of CF patients during MucoFong project.

**Figure 2 jcm-10-03725-f002:**
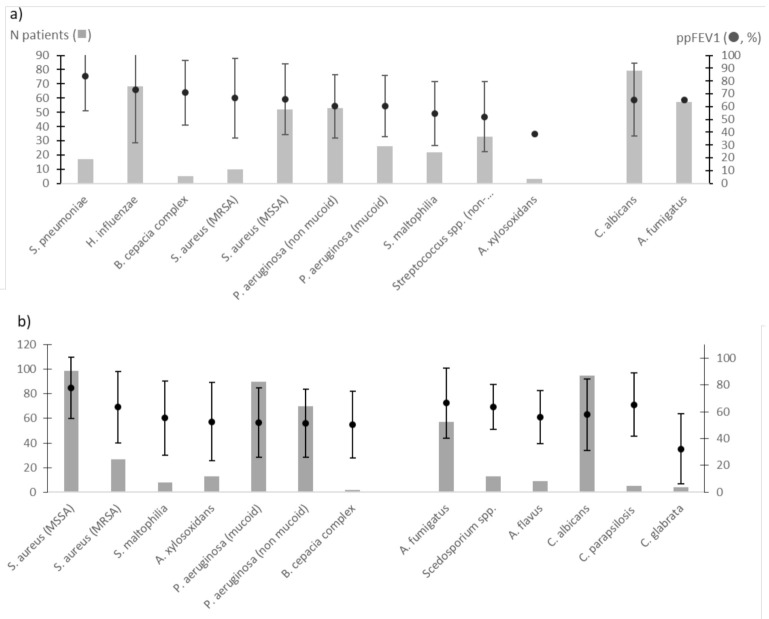
Mean ppFEV1 and its 95% confidence interval of patients with microbial transient (**a**) and chronic (**b**) colonisations at inclusion.

**Table 1 jcm-10-03725-t001:** Characteristics of patients at inclusion (*n* = 299).

Characteristics	Missing Data	*n* or Median or Mean	% or [IQR] or (SD)
Age (years)	0	23.7	(10.9)
Gender (female)	0	143	47.8
CFTR mutations	0		
F508del/F508del		119	39.8
F508del/other		136	45.5
Other/other or unknown		44	14.7
Comorbidities	0		
Exocrine pancreatic insufficiency		235	78.6
Diabetes mellitus		83	27.8
Gastroesophageal reflux		73	24.4
ABPA		69	23.1
Chronic sinusitis		58	19.4
Haemoptysis		55	18.4
Concomitant asthma		34	11.4
Pneumothorax		9	3.0
Pulmonary hypertension		5	1.7
BMI (kg/m^2^)	1	19.9	[17.6–21.3]
Shwachman–Kulczycki score	28	74.6	[65–90]
Number of days of hospitalization during the last 12 months *	10	7	[1–115]
Sputum bacterial cultures positive with			
*Staphylococcus aureus* (methicillin susceptible)		80	27.4
*Pseudomonas aeruginosa* (non-mucoid strains)		75	26.4
*Pseudomonas aeruginosa* (mucoid strains)		77	22.6
*Haemophilus influenzae*		28	9.6
*Staphylococcus aureus* (methicillin resistant)		22	7.5
Non-pneumoniae *Streptococcus* species		19	6.5
*Stenotrophomonas maltophilia*		16	5.5
*Alkaligenes xylosoxidans*		12	4.1
*Streptococcus pneumoniae*		5	1.7
*Bulkhoderia cepatia* complex		3	1.0
*Mycobacterium avium complex*		4	1.4
*Mycobacterium abscessus complex*		5	1.7
Sputum fungal cultures positive with	10		
*Candida albicans*		136	45.5
*Aspergillus fumigatus*		80	26.8
Other moulds *(Cladosporium* sp., *Penicillium* sp.)		20	6.7
*Scedosporium* species		10	4.0
*Aspergillus flavus*		9	3.0
*Candida parapsilosis*		5	1.7
Other *Aspergillus* species		4	1.3
*Candida glabrata*		3	1.0
Other *Candida* species		3	1.0
*Exophiala dermatitidis*		2	0.7
*Lomentospora prolificans*		1	0.3
Medications	0		
Non-invasive ventilation		21	7.2
Long-term oxygen		32	10.7
Nebulized rhDNase		178	59.5
Inhaled antibiotics		86	28.8
Inhaled steroids		168	56.2
Azithromycin		162	54.2
Oral antibiotic courses during the last 12 months			
0		75	25.1
1 to 3		161	53.8
>3		63	21.1
Intravenous antibiotics during the last 12 months		153	51.2
Mean number of antibiotic courses		1	[0–2]
Systemic corticosteroids during the last 12 months		70	23.4
Other immunosuppressive treatment		3	1.0
Antifungal treatment during the last 6 months		61	20.4

ABPA, allergic bronchopulmonary aspergillosis; BMI, body mass index. Antifungal treatment was: itraconazole in 48, voriconazole in 14, fluconazole in 4, and posaconazole in 5 of the cases. * Excluding patients without hospitalization.

**Table 2 jcm-10-03725-t002:** ppFEV1 predicted values and health care utilization during the 2 years of follow-up.

Variables Expressed as Median [IQR] or Mean (±SD)—(MV)	Inclusion	1st Follow-Up (Year 1)	2nd Follow-Up (Year 2)
ppFEV1	67 [43–91]—(4)	70 [41–92]—(36)	69 [47–91]—(68)
Outpatient visits (n)	4 [3–6]—(1)	4 [3–5]—(5)	4 [1–5]—(13)
Hospital admissions (n)	0 [0–1]—(1)	0 [0–2]—(6)	0 [0–1]—(13)
Days of hospitalization (n)	12.2 (±16.6)—(10)	13.0 (±18.3)—(59)	12.7(±16.8)—(22)

Data are presented as median with interquartile range [IQR] or mean and standard deviation (± SD) followed by missing values (MV); ppFEV1, percent predicted forced expiratory volume in 1 s; numbers of consultations (hospital admissions) and hospitalizations recorded during the last 12 months before the visit.

**Table 3 jcm-10-03725-t003:** Microbial factors at the inclusion associated with the average change of ppFEV1 during the 2 years of follow-up based on a univariate analysis.

Microorganisms	Type of Colonisation (Number of Patients)	β Coefficient	95%CI	*p*-Value
*H. influenzae*	Transient (68)	5.23	[1.47; 11.94]	0.13
*S. pneumoniae*	Transient (17)	14.79	[3.47; 26.11]	0.01
non-pneumoniae *Streptococcus*	Transient (33)	−6.93	[−18.95; 5.09]	0.25
*S. aureus* (methicillin susceptible)	Transient (53)	4.60	[−2.84; 12.04]	0.17
	Chronic (99)	5.69	[−0.71; 12.10]	
*S. aureus* (methicillin resistant)	Transient (10)	1.60	[−13.09; 16.28]	0.38
	Chronic (27)	−6.44	[−15.72; 2.84	
*P. aeruginosa* (non-mucoid strains)	Transient (53)	−9.77	[−16.78; −2.76]	<0.0001
	Chronic (70)	−15.41	[−22.08; −8.75]	
*P. aeruginosa* (mucoid strains)	Transient (26)	−19.45	[−28.74; −10.16]	<0.0001
	Chronic (90)	−13.15	[−19.42; −6.89]	
*S. maltophilia*	Transient (22)	−14.19	[−24.22; −4.16]	0.01
	Chronic (8)	−10.26	[−26.44; 5.93]	
*B. cepacia* complex	Transient (5)	18.60	[−2.02; 39.23]	0.15
	Chronic (2)	−13.15	[−44.69; 18.40]	
*A. xylosoxidans*	Transient (3)	−6.32	[−32.99; 20.35]	0,03
	Chronic (13)	−17.97	[−31.32; −4.62]	
*A. fumigatus*	Transient (57)	−2.99	[−10.05; 4.07]	0.69
	Chronic (57)	−1.53	[−8.65; 5.60]	
*Scedosporium species*	Transient (0)	/	/	0.22
	Chronic (10)	−7.42	[−22.56; 7.52]	
*C. albicans*	Transient (79)	−4.74	[−11.23; 1.74]	
	Chronic (85)	−11.03	[−17.62; −4.44]	0.005

**Table 4 jcm-10-03725-t004:** Microbial factors at inclusion associated with the average change of ppFEV1 during the 2-year follow-up in the mixed linear model.

Microorganisms	Type of Colonisation (Number of Patients)	β Coefficient	95%CI	*p*-Value
*S. pneumoniae*	Transient (17)	10.92	0.45; 21.40	0.04
non-pneumoniae *Streptococcus* species	Transient (33)	−13.62	−25.12; −2.12	0.02
*A. xylosoxidans*	Transient (3)	−3.61	−26.95; −2.30	0.06
	Chronic (13)	−14.62	−32.08; 24.86	
*S. maltophilia*	Transient (22)	−11.93	−21.38; −2.49	0.02
	Chronic (8)	−11.02	−26.83; 4.79	
*P. aeruginosa* (non-mucoid strains)	Transient (53)	−7.54	−14.30; −0.79	0.01
	Chronic (70)	−9.63	−16.61; −2.65	
*P. aeruginosa* (mucoid strains)	Transient (26)	−18.26	−27.20; −9.31	<0.0001
	Chronic (90)	−10.41	−16.96; −3.87	
*C. albicans*	Transient (79)	−5.12	−11.17; 0.93	<0.01
	Chronic (85)	−10.26	−16.50; −4.02	

**Table 5 jcm-10-03725-t005:** Microbial factors associated with the prognostic score for the prediction of within 3-years death or lung transplantation estimated at inclusion.

Microorganisms	Type of Colonisation (Number of Patients)	β Coefficient	95%CI	*p*-Value
*S. pneumoniae*	Transient (17)	−0.05	−0.81; 0.71	0.90
non-pneumoniae *Streptococcus* spp.	Transient (33)	0.53	−0.23; 1.30	0.17
*A. xylosoxidans*	Transient (3)	0.49	0.84; 2.59	0.001
	Chronic (13)	1.72	−1.27; 2.25	
*S. maltophilia*	Transient (22)	0.90	0.21; 1.55	0.01
	Chronic (8)	0.88	−0.18; 1.99	
*P. aeruginosa* (non-mucoid strains)	Transient (53)	0.95	0.48; 1.43	<0.0001
	Chronic (70)	1.30	0.80; 1.81	
*P. aeruginosa* (mucoid strains)	Transient (26)	0.91	0.26; 1.56	0.01
	Chronic (90)	0.41	−0.07; 0.88	
*C. albicans*	Transient (79)	0.21	−0.23; 0.65	0.63
	Chronic (85)	0.12	−0.32; 0.56	

## Data Availability

The data that support the findings of this study are available from the corresponding author (L.D.), upon reasonable request.

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
