# Peer review of "New Insights in Microbial Species Predicting Lung Function Decline in CF: Lessons from the MucoFong Project"

_jcm, 2021, doi:10.3390/jcm10163725_

Round 1

Reviewer 1 Report

This study used a traditional culture-based approach to track bacterial and fungal infection/colonization over a two-year period in a cohort of 299 French cystic fibrosis patients. Using pulmonary function measurements (ppFEV1) and a lung transplant/ death prognostic score the authors generate univariate and multivariate statistical models to analyze associations between different organisms with change in lung function and disease outcome. In these models they adjust for other clinical parameters, including patient age, sex, BMI, and several other variables. They find that P. aeruginosa, S. maltophila, A. xylosoxidans, C. albicans, and non-pneumoniae Streptococcus spp. were associated with a decrease in lung function in a mixed linear model. Likewise, these same species were also associated with higher likelihood of death or lung transplant; however, the contribution of non-pneumoniae Streptococcus spp. was not significant (p=0.17). The strengths of this study include the relatively large patient cohort, the focus on fungal colonization which has been understudied CF relative to bacterial colonization, the wealth of patient data and associated microbial culture data, and the statistical modeling. The weaknesses of this paper are missing details in the methods section, the failure to account for polymicrobial infections, and the potential overinterpretation of the role of non-pneumoniae Streptococcus spp. in patient outcomes in this study. Overall, this study adds to an ever-growing body of literature on the roles of microbes in CF lung disease progression and with minor changes as well as one potential new analysis of the existing data, could build on this further.

Major critiques and comments:

  • The authors focus heavily on the role of non-pneumoniae Streptococcus spp. in the CF lung disease progression. Yet, their data show that other organisms are clearly more relevant to disease. For example, in the univariate models non-pneumoniae Streptococcus spp. were not significantly associated with a decrease in ppFEV1 (p=0.25) and the change in ppFEV1 (-6.93) was less than the change associated with transient mucoid aeruginosa (-19.45, p<0.0001), transient S. maltophila (‑14.19, p=0.01), chronic A. xylosoxidans (‑17.97, p=0.03), and chronic C. albicans (-11.03, p=0.005). The same was true in the ppFEV1 multivariate model where transient mucoid P. aeruginosa had the greatest negative association with ppFEV1, and in the multivariate prognostic model the association of non-pneumoniae Streptococcus spp. with poor outcome was non-significant (p=0.17). Together, these data highlight that traditional CF pathogens remain the best predictors of disease outcomes in adults with CF, and this should be the primary conclusion in the discussion. If the authors want to discuss the role of non-pneumoniae Streptococcus spp. they might consider the roles of these organisms in relation to other pathogens like P. aeruginosa, see previous studies from Jorth et al. 2020. Cell Reports and Zhao et al. 2012. Proc. Natl. Acad. Sci. and Sibley et al. 2008. Proc. Natl. Acad. Sci.

  • The authors discuss the potential importance polymicrobial interactions; however, they do not analyze the polymicrobial nature of the samples collected for this study. For example, how often were different organisms detected in the same samples (i.e., aeruginosa and S. aureus which are the most common co-infecting organisms, or P. aeruginosa or other pathogens with non-pneumoniae Streptococcus spp.). Could this analysis be included? If not, this ought to be listed as a limitation.

  • A limitation to this study is that the microbes were not quantified. Recent quantitative studies have shown that oral organisms like non-pneumoniae Streptococcus spp. are more commonly detected in samples where bacterial loads are high and pathogens like aeruginosa or S. aureus are highly abundant. While it is likely unrealistic to quantify bacterial and fungal burdens in these sputum samples, it should be noted in the study limitations that this is an important parameter that was not accounted for.

  • Methods: How were bacteria and fungi cultured and identified? The authors reference the European CF Society guidelines, but a more detailed description is needed to fully evaluate the study. Also, please list the semi-selective media which were used, and how growth on each type was interpreted.

  • Methods: Please describe the prognostic score assessment in more detail.

Minor comments:

Throughout the manuscript the authors refer to changes in ppFEV1 as “evolution” of ppFEV1. It suggested that they replace “evolution” with “change”, “increase”, or “decrease”. The term evolution could be misinterpreted to mean Darwinian evolution, which is inaccurate. Line 19, suggest changing “CF evolution” to “CF disease progression”. See also lines 21, 112. 114, 133, 212, 230, 226, and 265.

Line 47: “(LTLT)” should be “(LT)”.

Lines 47-49: Please re-phrase this sentence. What is meant by, “provided data that might be useful for daily practice but need to be handled by other clinicians from other CF centers.”? Do you mean that other centers need to generate data? Or do other centers need to interpret the data that was generated in these previous studies? Please clarify.

Lines 138-141: It is stated that interactions between non-pneumoniae Streptococcus spp. and C. albicans would be explored, yet this interaction is not specifically analyzed in the results. Suggest removing this statement.

Line 202: Please clarify, for univariate analyses were ppFEV1 from patients infected/colonized with a given organism compared to all other patients, or compared to patients that had zero colonizing/infecting organisms?

Line 220-221: Suggest re-phrasing to “Furthermore, transient colonization with non-pneumoniae Streptococcus spp. was associated with a 13.6% decrease in ppFEV1…”.

Line 239: Suggest re-phrasing to include non-pneumoniae Streptococcus, “With the exception of C. albicans and non-pneumoniae Streptococcus spp., ”.

Line 240: Suggest removing “non-pneumoniae Streptococcus” from the list of species significantly associated with prognostic score, because p=0.17.

Lines 288-291: Suggest removing this paragraph, or acknowledging that non-pneumoniae Streptococcus were only significantly associated with decreased ppFEV1 in one of the three models analyzed.

Table 1/General question: Were any patients on CFTR corrector therapies in this study? If so, the authors should consider correcting for this in their statistical models or acknowledge the limitation of not including this important therapy in the discussion.

Fig. 2: Please include more details in the legend. Is this the baseline ppFEV1 value? What is being plotted for ppFEV1, mean or median? Are these values for those in which the specific organisms were detected? What do the error bars indicate? Please change the y-axes to remove lines that go across the graphs, instead show axis lines with tick marks at right and left. Currently the axis lines correspond to number of patients, but they can be confused with the ppFEV1 which is plotted at right.

Lines 198-201, Fig. 2: How was baseline ppFEV1 determined to be lower for the species listed? Based on the overlapping error bars in Fig. 2, it appears that there would not be any significant differences in ppFEV1 for patients infected with/colonized by any of the given species at baseline.

Table 3, 4, 5: For organisms that have both transient and chronic colonization, why is only one p-value shown? This suggests that they were not analyzed individually.

Table 3, 4, 5: Please provide an additional column for the number of patients that were transiently or chronically colonized by each organism.

Table 5: Please provide more descriptive title, suggest adding time to death or lung transplant.

Author Response

Dear Editor of the Journal of Clinical Medicine

Dear Reviewers

Please find enclosed the revised manuscript “New Insights in Microbial Species Predicting Lung Function Decline in CF: Lessons from the MucoFong Project” (Manuscript ID: jcm-1287774) that was changed according to the recommendations addressed by the reviewers and the editorial office.

Please find below the point-by-point responses (in blue) to the Editor and Reviewer’s comments:

COMMENTS FROM REVIEWER #1:

MAJOR COMMENTS:

1) The authors focus heavily on the role of non-pneumoniae Streptococcus spp. in the CF lung disease progression. Yet, their data show that other organisms are clearly more relevant to disease. For example, in the univariate models non-pneumoniae Streptococcus spp. were not significantly associated with a decrease in ppFEV1 (p=0.25) and the change in ppFEV1 (-6.93) was less than the change associated with transient mucoid aeruginosa (-19.45, p<0.0001), transient S. maltophila (‑14.19, p=0.01), chronic A. xylosoxidans (‑17.97, p=0.03), and chronic C. albicans (-11.03, p=0.005). The same was true in the ppFEV1 multivariate model where transient mucoid P. aeruginosa had the greatest negative association with ppFEV1, and in the multivariate prognostic model the association of non-pneumoniae Streptococcus spp. with poor outcome was non-significant (p=0.17). Together, these data highlight that traditional CF pathogens remain the best predictors of disease outcomes in adults with CF, and this should be the primary conclusion in the discussion. If the authors want to discuss the role of non-pneumoniae Streptococcus spp. they might consider the roles of these organisms in relation to other pathogens like P. aeruginosa, see previous studies from Jorth et al. 2020. Cell Reports and Zhao et al. 2012. Proc. Natl. Acad. Sci. and Sibley et al. 2008. Proc. Natl. Acad. Sci.

RESPONSE: The authors agreed and amended the first sentence of the discussion section as follows: “Our results were in agreement with previous findings indicating that several concomitant chronic bacterial colonisations at the time of diagnosis (P. aeruginosa, B. cepacia and S. maltophilia) confer worse ppFEV1 level in CF; traditional CF pathogens remaining the best predictors of disease outcomes in adults with CF [3,5,6,10,12,14,16,17,26–33]“ (see lines 289-292).  

The roles of non-pneumoniae Streptococcus spp. in relation to P. aeruginosa has been discussed, as follows (Lines 303-313): “Moreover, a longitudinal analysis of the bacterial composition and dynamics of sputum samples from a CF adult patient chronically colonized by P. aeruginosa revealed the key role of Streptococcus milleri group in establishing chronic lung disease but also in exacerbation initiations [38]. These data highlight that considering the mechanisms of pathogenicity is complex and relied on the polymicrobial context of CF airways, in which P. aeruginosa is a recognized major pathogen interacting with the other members of the CF lung microbial community including non-pneumoniae Streptococcus, which may shape P. aeruginosa pathogenicity [38-41]”. The references (Jorth et al. 2020. Cell Reports and Zhao et al. 2012. Proc. Natl. Acad. Sci. and Sibley et al. 2008. Proc. Natl. Acad. Sci. & O'Brien, S. et al. FEMS microbiology letters, 2017) have been added.

2) The authors discuss the potential importance polymicrobial interactions; however, they do not analyze the polymicrobial nature of the samples collected for this study. For example, how often were different organisms detected in the same samples (i.e., aeruginosa and S. aureus which are the most common co-infecting organisms, or P. aeruginosa or other pathogens with non-pneumoniae Streptococcus spp.). Could this analysis be included? If not, this ought to be listed as a limitation.

RESPONSE: Regarding the polymicrobial infections, we explored several coinfections (Non-mucoide or mucoide strains of P. aeruginosa in combination with either S. aureus, non-pneumoniae Streptococcus species, A. fumigatus, Scedosporium sp., C. albicans, or C. glabrata) using univariate analysis that didn’t reveal any significant impact on ppFEV1. This point was mentioned in the results section (Lines 234-239).

Coinfection results focused on P. aeruginosa have been summarized lines 192-202 as follows: “As previously described [1,19], the number of fungal species isolated was increased with patient age: from none (18.9%) to 1 (42%), 2 (29.6%), 3 (8.2%), 4 (0.8%) and 5 (0.4%) species. In addition, patients colonized with P. aeruginosa were mainly colo-nized with two or more microbial species: 33 of the 75 patients colonized with non-mucoid P. aeruginosa were co-infected only with one of the following species mu-coid P. aeruginosa (n=7), non-pneumoniae Streptococcus (n=4), SAMS (n=11), SAMR (n=2), C. albicans (n=4) or A. fumigatus (n=5) species, and 29 of the 77 patients colonized with P. aeruginosa (mucoid strains) were co-infected only with one of the following species non-mucoid P. aeruginosa (n=7), non-pneumoniae Streptococcus (n=2), SAMS (n=3), SAMR (n=1), C. albicans (n=7) or A. fumigatus (n=9) species”.

3) A limitation to this study is that the microbes were not quantified. Recent quantitative studies have shown that oral organisms like non-pneumoniae Streptococcus spp. are more commonly detected in samples where bacterial loads are high and pathogens like aeruginosa or S. aureus are highly abundant. While it is likely unrealistic to quantify bacterial and fungal burdens in these sputum samples, it should be noted in the study limitations that this is an important parameter that was not accounted for.

RESPONSE: This limitation has been added line 344.

4) Methods: How were bacteria and fungi cultured and identified? The authors reference the European CF Society guidelines, but a more detailed description is needed to fully evaluate the study. Also, please list the semi-selective media which were used, and how growth on each type was interpreted.

RESPONSE: The semi-selective media have been detailed lines 73-86, as follows:” Briefly, in agreement with the French guidelines, 10 µL of pure sputum sample were plated on a selective agar for Gram-negative bacilli isolation (such as MacConkey type), a selective agar for P. aeruginosa isolation (optional if the previous selective agar allows accurate identification of P. aeruginosa colonies), a selective agar for B. cepacia complex isolation (such as PC or BCSA), and a selective agar for S. aureus isolation including small colony variants (e.g. hypersaline mannitol agar, chromogenic media). These media were incubated at 35 ± 2°C in an aerobic atmosphere for selective media and in a CO2-enriched atmosphere for non-selective media. A minimum detection limit equal to 102 CFU/mL will be used to interpret growth on each type of media. In parallel, 10 µL of sputum sample diluted to 1/1000 (detection threshold equal to 105 CFU/mL) were plated on an agar allowing the growth of S. pneumoniae and S. aureus with inhibition of Gram-negative bacteria (e.g. fresh blood ANC, CAP), and a cooked blood agar with or without bacitracin and with growth factors for the cultivation of H. influenza. These two media were incubated at 35 ± 2°C in a CO2-enriched atmosphere.”

5) Methods: Please describe the prognostic score assessment in more detail.

RESPONSE: Lines 115-117, the prognostic score was more detailed, as follows: “The recently proposed risk score for the prediction of within 3-year death or lung transplantation (LT) [12] was estimated for each patient at inclusion. Based on the 8 predictors proposed (i.e., ppFEV1, BMI, B. cepacia complex colonization, number of in-travenous antibiotics courses per year, number of days of hospitalization per year, oral corticosteroids therapy, long-term oxygen therapy, and non-invasive ventilation), a weighted continuous score was individually calculated, a high prognostic score corre-sponding to a high risk of LT or death.“

MINOR COMMENTS:

1) Throughout the manuscript the authors refer to changes in ppFEV1 as “evolution” of ppFEV1. It suggested that they replace “evolution” with “change”, “increase”, or “decrease”. The term evolution could be misinterpreted to mean Darwinian evolution, which is inaccurate. Line 19, suggest changing “CF evolution” to “CF disease progression”. See also lines 21, 112. 114, 133, 212, 230, 226, and 265.

RESPONSE: The authors have replaced in the whole article “evolution” with progression, “change”, “increase”, or “decrease” depending on the context.  

2) Line 47: “(LTLT)” should be “(LT)”. RESPONSE: Modification done

3) Lines 47-49: Please re-phrase this sentence. What is meant by, “provided data that might be useful for daily practice but need to be handled by other clinicians from other CF centers.”? Do you mean that other centers need to generate data? Or do other centers need to interpret the data that was generated in these previous studies? Please clarify. RESPONSE: The sentence has been re-phrased. 

4) Lines 138-141: It is stated that interactions between non-pneumoniae Streptococcus spp. and C. albicans would be explored, yet this interaction is not specifically analyzed in the results. Suggest removing this statement. RESPONSE: The sentence has been deleted.

5) Line 202: Please clarify, for univariate analyses were ppFEV1 from patients infected/colonized with a given organism compared to all other patients, or compared to patients that had zero colonizing/infecting organisms?

RESPONSE: The authors clarified the method as follows (lines 137-141): “According to the unadjusted univariate analyses, potential microbial predictors of the outcome such as transient or chronic bacterial and fungal colonizations (yes/no) were initially introduced in the model. Univariate analysis compared ppFEV1 from patients infected/colonized with a given organism compared to all other patients.”

6) Line 220-221: Suggest re-phrasing to “Furthermore, transient colonization with non-pneumoniae Streptococcus spp. was associated with a 13.6% decrease in ppFEV1…”.

RESPONSE: The sentence was re-phrased (lines 254-256).

7) Line 239: Suggest re-phrasing to include non-pneumoniae Streptococcus, “With the exception of C. albicans and non-pneumoniae Streptococcus spp., ”.

RESPONSE: The sentence was re-phrased (lines 273-275).

8) Line 240: Suggest removing “non-pneumoniae Streptococcus” from the list of species significantly associated with prognostic score, because p=0.17.

RESPONSE: The sentence was re-phrased (lines 273-277).

9) Lines 288-291: Suggest removing this paragraph, or acknowledging that non-pneumoniae Streptococcus were only significantly associated with decreased ppFEV1 in one of the three models analyzed.

RESPONSE: The sentence has been modified as follow: “On the opposite, non-pneumoniae Streptococcus carriage was correlated with decreased ppFEV1 in one of the three models analysed a negative evolution of ppFEV1 in our population” (lines 302-304).

10) Table 1/General question: Were any patients on CFTR corrector therapies in this study? If so, the authors should consider correcting for this in their statistical models or acknowledge the limitation of not including this important therapy in the discussion.

RESPONSE: None of the patients were under CFTR corrector.

11) Fig. 2: Please include more details in the legend. Is this the baseline ppFEV1 value? What is being plotted for ppFEV1, mean or median? Are these values for those in which the specific organisms were detected? What do the error bars indicate? Please change the y-axes to remove lines that go across the graphs, instead show axis lines with tick marks at right and left. Currently the axis lines correspond to number of patients, but they can be confused with the ppFEV1 which is plotted at right.

RESPONSE: The Figure 2 has been revised.

12) Lines 198-201, Fig. 2: How was baseline ppFEV1 determined to be lower for the species listed? Based on the overlapping error bars in Fig. 2, it appears that there would not be any significant differences in ppFEV1 for patients infected with/colonized by any of the given species at baseline.

RESPONSE: The author used a linear mixed model to evaluate the effect of transient or chronic colonizations with fungi and/or bacteria at the inclusion on the changes of ppFEV1. The model included a random intercept; parameters were estimated with the maximum likelihood method. A graphical analysis of the data revealed a heterogeneous change of ppFEV1 values among patients throughout the follow-up. It was therefore decided to consider time as a discrete variable (one variable per visit). No additional time interaction was required in the model; therefore, the effects of bacteria or fungi were modelled as an average difference of ppFEV1 from the inclusion onward (as mentioned in the material and method section).

13) Table 3, 4, 5: For organisms that have both transient and chronic colonization, why is only one p-value shown? This suggests that they were not analyzed individually.

RESPONSE: Each type of colonization was analyzed individually but the value represents the p-value of the overall change since the effect of a given microorganism on the ppFEV1 change seems to be in the same direction for both types of colonization.

14) Table 3, 4, 5: Please provide an additional column for the number of patients that were transiently or chronically colonized by each organism.

RESPONSE: The numbers of patients that were transiently or chronically colonized by each organism have been added in tables 3-5.

15) Table 5: Please provide more descriptive title, suggest adding time to death or lung transplant.

RESPONSE: The tile of Table 5 has been modified as follows: “Microbial factors associated with the prognostic score for the prediction of within 3-year death or lung transplantation estimated at inclusion”.

To complete the revision, the authors updated the references accordingly to the reviewer’s comments.

Our revision includes:

-(1) a letter with point-by-point responses to the reviewer comments (file: Point-by-point responses_Francis et al),

-(2) a marked revision of our manuscript showing the changes made (under revision mode in Word, file: 20210806_jcm-1287774_revised version), and

-(3) the revised version of the figure 2 (Figure 2 revised version).

Sincerely yours,

Pr Laurence Delhaes

Reviewer 2 Report

Knowledge of the natural history of CF-associated lung disease has been (and still is) the subject of study for many years, particularly those aspects related to survival and referral for lung transplantation. In this manuscript, the authors evaluate in 299 patients with CF the association between the presence in sputum of several bacterial and fungal species and the evolution of lung function as determined by the evolution of FEV1% over 2 years. The authors confirm previous results of other studies and point out the importance of knowing the complete microbiota colonizing the lungs of CF patients for a better understanding of the pulmonary evolution of these patients.

The article is interesting from a clinical point of view, although it presents some aspects that could be improved.

My main concern is the method used for outcome assessment. The authors use spirometry (FEV1%) to assess lung function. Although FEV1 is an established surrogate for assessing survival in CF patients and has been widely used to identify the stage of lung disease in CF patients, it has not been found to predict lung function decline in many studies. The authors should discuss this point, explain why only FEV1% is used and why other spirometry parameters such as FVC, FEF25-75, etc., are not studied. The question to be answered is whether the inclusion of these parameters better differentiates between patients who will progress to end-stage lung disease.

In addition, since spirometry shows a lack of sensitivity to detect early structural abnormalities and progression in structural lung disease, other more sensitive measures for early identification of disease progression that have been used should also be discussed.

Author Response

Dear Editor of the Journal of Clinical Medicine

Dear Reviewers

Please find enclosed the revised manuscript “New Insights in Microbial Species Predicting Lung Function Decline in CF: Lessons from the MucoFong Project” (Manuscript ID: jcm-1287774) that was changed according to the recommendations addressed by the reviewers and the editorial office.

Please find below the point-by-point responses (in blue) to the Editor and Reviewer’s comments:

COMMENTS FROM REVIEWER #2:

Knowledge of the natural history of CF-associated lung disease has been (and still is) the subject of study for many years, particularly those aspects related to survival and referral for lung transplantation. In this manuscript, the authors evaluate in 299 patients with CF the association between the presence in sputum of several bacterial and fungal species and the evolution of lung function as determined by the evolution of FEV1% over 2 years. The authors confirm previous results of other studies and point out the importance of knowing the complete microbiota colonizing the lungs of CF patients for a better understanding of the pulmonary evolution of these patients.

The article is interesting from a clinical point of view, although it presents some aspects that could be improved.

My main concern is the method used for outcome assessment. The authors use spirometry (FEV1%) to assess lung function. Although FEV1 is an established surrogate for assessing survival in CF patients and has been widely used to identify the stage of lung disease in CF patients, it has not been found to predict lung function decline in many studies. The authors should discuss this point, explain why only FEV1% is used and why other spirometry parameters such as FVC, FEF25-75, etc., are not studied. The question to be answered is whether the inclusion of these parameters better differentiates between patients who will progress to end-stage lung disease.

In addition, since spirometry shows a lack of sensitivity to detect early structural abnormalities and progression in structural lung disease, other more sensitive measures for early identification of disease progression that have been used should also be discussed.

RESPONSE: The authors agreed; they added several sentences to explain and discuss the use of ppFEV1, in lines 337 -343. 

To complete the revision, the authors updated the references accordingly to the reviewer’s comments.

Our revision includes:

-(1) a letter with point-by-point responses to the reviewer comments (file: Point-by-point responses_Francis et al),

-(2) a marked revision of our manuscript showing the changes made (under revision mode in Word, file: 20210806_jcm-1287774_revised version), and

-(3) the revised version of the figure 2 (Figure 2 revised version).

Sincerely yours,

Pr Laurence Delhaes

Round 2

Reviewer 1 Report

The author's have done an excellent job responding to the critiques.